# Quality of life following a lower limb reconstructive procedure: a protocol for the development of a conceptual framework

Heather Leggett ![ORCID],[1] Arabella Scantlebury ![ORCID],[1] Hemant Sharma,[2] Catherine Hewitt,[1] Melissa Harden,[3] Catriona McDaid ![ORCID],[1] On behalf of the PROLLIT study collaborators

► Prepublication history and additional materials for this paper is available online. To view these files, please visit the journal online (http://dx.doi.org/10.1136/bmjopen-2020-040378).

¹York Trials Unit, University of York, York, UK
²Trauma and Orthopedics, Hull Royal Infirmary, Hull, UK
³Centre for Reviews and Dissemination (CRD), University of York, York, UK

**Correspondence to**
Heather Leggett;
heather.leggett@york.ac.uk

## ABSTRACT

**Introduction** Lower limb conditions requiring reconstructive surgery can be either congenital or acquired from trauma, infection or other medical conditions. Patient-reported outcome measures (PROMs) are often used by healthcare professionals to assess the impact of a patient's condition (and treatment) on quality of life. However, we are not aware of any measures developed specifically for people requiring lower limb reconstructive surgery. Consequently, it is not clear the extent to which current PROMs accurately and specifically measure the outcomes that are important to these patients.

**Methods and analysis** The 'PROLLIT' (Patient-Reported Outcome Measure for Lower Limb Reconstruction) involves three phases: to explore what is important to patients with regard to quality of life (phase 1), ascertain whether current measures adequately capture these experiences (phase 2) and if not begin, the development of a new PROM (phase 3). The population of interest is people requiring, undergoing or after undergoing reconstructive surgery for a lower limb condition. In this paper, we describe phase 1, which aims to develop a conceptual framework to identify and map what is important to this group with regard to social interactions, employment, perceived health and quality of life after condition onset/injury and throughout recovery. The conceptual framework will be developed through three steps: (step A) a qualitative evidence synthesis, (step B) a qualitative study with patients and staff to explore patient's views and experiences of lower limb reconstructive surgery and (step C) a round table discussion with key stakeholders where findings from step A and step B will be brought together and used to finalise the conceptual framework.

**Ethics consideration and dissemination** Ethical approval has been granted for the qualitative data collection (step B) from South Central Berkshire Research Ethics committee (REF:20/SC/0114). Findings from steps A and B will be submitted for peer-reviewed publication in academic journals, and presented at academic conferences.

**PROSPERO registration number** CRD42019139587.
**ISRCTN registration number** ISRCTN75201623.

### Strengths and limitations of this study

► A conceptual framework will be developed to establish the areas of importance for patients' quality of life and will determine if there is a need to develop a new patient-reported outcome measure.
► A multimethods approach will be used to develop the conceptual framework, increasing the robustness of the study.
► Both patient and staff opinions on important areas relating to quality of life after a lower limb reconstruction will be sought which will offer a more encompassing and detailed exploration.
► Due to resource constraints, only English language studies will be eligible for inclusion in the review, introducing the risk of language bias.
► There is a risk that some individuals may be more likely than others to agree to take part in the qualitative study limiting generalisability, though strategies are in place to mitigate against this.

## INTRODUCTION

There are several different lower limb conditions requiring reconstructive surgery. These can be congenital or acquired from trauma, infection or other medical conditions. Conditions may include, but are not limited to bone loss, joint contracture, non-union, malunion or bone deformity. Limb reconstruction is the orthopaedic subspecialty, treating sequel of complex trauma and congenital conditions; deformity correction, non-union, malunion, bone infections, limb lengthening and complex trauma. Individuals with a lower limb condition often experience mobility impairments as well as pain, discomfort, sleep disturbance, anxiety and low mood which can affect their daily life and reduce their employment and social engagement opportunities.[1] Therefore, it is not surprising that the experience of living with a lower limb condition as well as the impact of initial trauma or condition onset and any follow-on consequences may have negative impacts on an individual's quality of life (QOL).[2] Consequently, it is important to understand the experience of patients during and after reconstructive surgery for a lower limb condition in order to measure treatment effectiveness and to monitor and improve care, care

management, resource management and health policy. Healthcare professionals and researchers regularly use patient-reported outcome measures (PROMs) with patients. They often include assessments of a patient's specific condition and the impact (of the condition and treatment) on QOL.[3][4] These are important aspects for knowing if, and how to improve quality of care for patients.[4]

Commonly used outcome measures for this population are related to musculoskeletal function such as the Olerud-Molander Ankle Score,[5] or generic measures such as the Disability Rating Index,[6] Short Form 12 Health Survey,[7] The Short Form 36 Health Survey,[8] EQ-5D-5L[9] or the Sickness Impact Profile.[10] These tools have been used in capturing and reporting the QOL of patients who have experienced lower limb reconstruction.[11][12] More recently, the Patient-Reported Outcomes Measurement Information System Physical Function measure was validated in patients with an orthopaedic trauma to a lower extremity.[12][13] However, this measure did not include QOL assessment and was limited to patients with an isolated lower extremity fracture only (with either surgery or non-operative closed treatment). Research is currently also underway to develop a PROM specifically for children and adolescents with lower limb deformities[14] as well as for adult patients undergoing a lower limb circular frame fixation.[15] While the work conducted by Antonios and colleagues[15] will be relevant to a subset of our population, it excludes a large number of patients undergoing other reconstructive treatments. The focus of existing PROMs means it is not clear whether they accurately and specifically measure the outcomes that are important to patients requiring or undergoing reconstructive surgery for a lower limb condition. As such, the need for a measure specifically designed to capture outcomes important to those requiring or undergoing reconstructive surgery for a lower limb condition may be warranted and needs further exploration.

Despite increased awareness surrounding the value of including patients in the development of PROMs, patients are not being fully engaged[16] and there is concern over the extent to which existing PROMs used with patients who have experienced a lower limb condition are fit for the purpose of accurately capturing important patient experiences. We are not aware of any measures developed specifically for people with conditions requiring reconstructive surgery. This study, known as 'PROLLIT' (Patient-Reported Outcome Measure for Lower Limb Reconstruction), will address this evidence gap by adopting a three-phase approach (figure 1) to explore what is important to patients with regard to QOL (phase 1); ascertain whether current measures adequately capture these experiences (phase 2) and if not, to begin the development of a new measure (phase 3).

In this protocol paper, we outline phase 1 of the PROLLIT study, which will involve developing a conceptual framework related to patients requiring, undergoing

**PHASE 1**
Develop conceptual framework
— Qualitative evidence synthesis and qualitative data collection to develop a conceptual framework of what is important to people requiring, undergoing or who have undergone reconstructive surgery for a lower limb condition.

**PHASE 2**
Establish need for new PROM
— Mapping the conceptual framework onto what existing PROMS measure to determine whether patient needs are currently met.

**PHASE 3**
If current PROMS do not meet patient need
— Develop first version of a PROM (Based on Phase 1 and with input from experts and patient-public engagement).

**Figure 1** The PROLLIT study. PROMs, patient-reported outcome measures.

or after undergoing reconstructive surgery for a lower limb condition (figure 2).

## METHODS
### Development of a conceptual framework
In phase 1, our three-stepped approach for developing a conceptual framework involves a qualitative evidence synthesis (QES) (step A), a qualitative study with health professionals and patients (step B) and a round table discussion with key stakeholders (step C; figure 2). The conceptual framework will be developed to identify and map what is important to patients requiring, undergoing or after undergoing reconstructive surgery for a lower limb condition with regard to social interactions, employment, perceived health and QOL after injury and throughout recovery.

For readability, we describe the process of developing our conceptual framework in three steps. However, it is important to recognise that this is not a linear study and that framework development will be on-going and iterative with each 'step' of the study influencing the other and contributing to the framework's overall development. For example, the findings of the QES (step A) will result in an initial framework. These findings will then be used to develop the topic guides for the qualitative study (step B), which, through semistructured interviews will draw on the direct experience of health professionals and patients to explore in more depth, and where appropriate add to the findings of the QES. In step C, the findings of the QES and qualitative study will then be combined and taken to a round table discussion with key stakeholders, the purpose of which will be to refine and finalise our conceptual framework (figure 2).

The conceptual framework will then be used to inform the wider PROLLIT study, where we will, depending on whether patient priorities are being met by existing PROMs, develop a new PROM for people requiring,

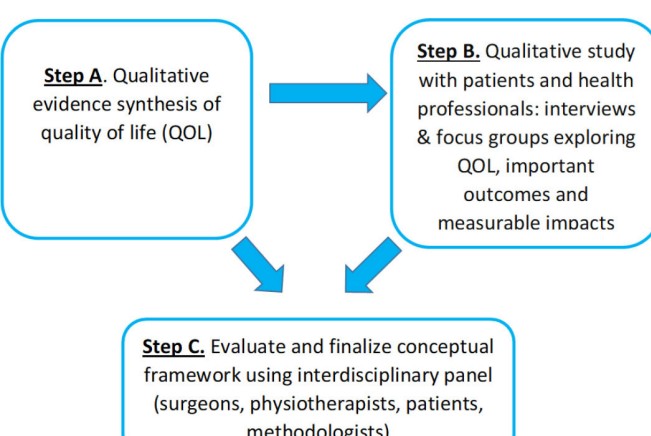

**Figure 2** The three phases of developing a conceptual framework for patients requiring, undergoing or after undergoing reconstructive surgery for a lower limb condition that are outlined in this paper.

undergoing or who have undergone reconstructive surgery for a lower limb condition (figure 1).

### Step A: Qualitative evidence synthesis

This QES will explore what is important to patients with regard to QOL after experiencing a lower limb reconstructive procedure. The Cochrane Handbook and guidance for QES published by the Cochrane Qualitative and Implementation Methods Group[17] provides the methodological framework for the design and implementation of this review. This QES will only include studies of 'good quality' based on the developed inclusion and exclusion criteria. The QES protocol complies with the requirements of the Preferred Reporting Items for Systematic Reviews and Meta-Analyses Protocol (PRISMA-P[18] as far as possible as PRISMA focuses on reviews of interventions. The QES findings (and final report) will be reported according to the enhancing transparency in reporting the synthesis of qualitative research (ENTREQ) guidelines,[19] as appropriate for qualitative evidence syntheses.

### Eligibility

#### Sample

Patients (adults: 16+) requiring, undergoing or who have undergone limb reconstructive surgery for a lower limb condition (leg, ankle or foot). Conditions may include the following:

► Infection: a fracture fixation which becomes infected.
► Non-union: a fracture which does not heal.
► Malunion/deformity: a fracture which does not heal in correct position. Any acquired or congenital condition leading to bone deformity.
► Leg length discrepancy or bone loss.
► Congenital lower limb deformities.
► Joint contracture.
► Lower limb injuries where further limb reconstruction is required.
► Poly-trauma patients (as long as one of the above criteria are met).

Patients who have undergone a lower limb amputation will not be included.

Time since condition onset/reconstructive surgery: patients will be included at any time point after condition onset. For example, participants will be included if they are still in hospital, under hospital outpatient care regarding their lower limb condition/reconstructive surgery or if they have been discharged with no further follow-up.

#### Phenomenon of interest

QOL, including (but not limited to) social interactions, employment, perceived health and QOL after condition onset/injury and throughout recovery.

#### Design

We will include qualitative studies which use established qualitative methods such as interviews or focus groups and use established qualitative analytical approaches (eg, thematic analysis, framework analysis, grounded theory) to analyse the findings. We will include mixed-methods research which meets the criteria of a qualitative study outlined above. We will exclude opinion pieces, commentaries, case studies, guidelines, audits, observational studies, randomised controlled trials and other experimental designs.

#### Evaluation

We will include studies reporting on patient's attitudes, perspectives and behaviours surrounding QOL in the broadest sense. This will encompass people's experiences of the condition (symptoms/pain/recovery), experiences of treatment, their physical, mental, emotional, social, daily and professional functioning (effect on working and any financial difficulties), as well as outcome expectations.

#### Research type

QES.

### Search strategy for the identification of relevant studies

The following databases will be searched: MEDLINE (including: Epub Ahead of Print, In-Process and Other Non-Indexed Citations, MEDLINE Daily and MEDLINE, via Ovid), Embase (Ovid), PsycINFO (Ovid) and CINAHL Complete (Ebsco). A preliminary search strategy has been developed for Ovid MEDLINE in conjunction with an Information Specialist from the Centre for Reviews and Dissemination at the University of York (online supplemental file 1). The strategy in brief consists of a set of terms for our population of interest combined using the Boolean operator AND, with a set of terms covering the research methods of interest (qualitative and mixed methods). The search terms relating to qualitative methods are based on previous search strategies evaluated by Shaw *et al*.[20] This strategy will be adapted for the remaining databases. Searches will be run from database inception until present, and will be limited to publications in English. A manual search of the reference list

of included studies will be undertaken to identify other studies meeting the inclusion criteria.

## Selection of studies and data extraction

All results will be imported into EndNote reference management software (Clarivate Analytics (formerly Thomson Reuters), Philadelphia, PA, USA) and deduplicated. Titles of records will be screened by a single researcher (task divided between three researchers) to remove obviously irrelevant records. Using the stated eligibility criteria, the abstract of each record included from the title screen stage will be independently reviewed by two researchers (task divided between three researchers). The full text of each record included from the abstract screen stage will then be independently screened by two researchers, against the eligibility criteria (task divided between three researchers). Discrepancies will be resolved by discussion with a third researcher.

Data extraction will be undertaken on the included studies using Excel for key study details. A data extraction template will be developed and piloted within the research team in order to summarise key information and persistent themes within studies. Key information will include author names, design or methods including method of data collection and analysis, participant characteristics and results. Results extracted will relate to patients' condition, treatments, outcomes, experiences, attitudes and thoughts surrounding QOL and other relevant outcomes using the NVIVO software. Data extraction will be undertaken by one researcher and checked by a second.

## Quality assessment

Guidance by the Cochrane Qualitative and Implementation Methods Group recommends researchers assess methodological strengths and weaknesses.[21] In line with this guidance, the Critical Appraisal Skills Programme[22] checklist will be used to assess the included studies. Each article will be assessed independently by two researchers. Any difficulties in identifying each study's methodological quality will be resolved by discussing the paper in question with a third researcher. GRADE-CERQual[23] will be used to assess the strength of the review's findings.

## Data analysis and synthesis

Data synthesis will be in line with the Cochrane Qualitative and Implementation Methods Group guidelines for QES.[24] Key study characteristics including details of the population, setting, methods and study quality will be tabulated and described narratively. Thematic synthesis will be undertaken to analyse the qualitative data; data analysis will be facilitated by the use of NVIVO. Thematic synthesis lends itself well to this research question and the type of data we anticipate finding since it allows for the synthesis and integration of both thin and more richly detailed data into descriptive and analytic themes.[25] The thematic synthesis will be based on the entirety of the 'results' section of each included study and will involve

a three-stage process.[26] First, both verbatim comments (quotes) and author observations (author interpretations) from included studies will be coded. Second, the identification of common findings from individual studies will lead to the creation of researcher developed descriptive themes to describe these findings. The first two stages will be undertaken independently by three researchers and then discussed as a group. Third, analytical themes will be developed to describe and interpret these themes, moving beyond the content of the original articles. For example, multiple instances of patient-reported frustration, fatigue, depression and anxiety may be grouped together as a theme entitled 'Psychological Impact' and then further explored to understand the impact of this on QOL. This is an iterative process achieved through independent and group discussions of the implications of each theme.

After our initial thematic analysis, we will hold a round table discussion among the research team to develop a framework where we will identify the important areas related to QOL for patients requiring, undergoing or after undergoing reconstructive surgery for a lower limb condition that were identified in the synthesis. These findings will be used to inform topic guides for the qualitative data collection in step B and will contribute to the development of the initial, hypothesised conceptual framework in step C (see figure 2).

## Step B: ualitative data collection
### Design

A qualitative study using semistructured interviews and focus groups will be undertaken with patients and staff (orthopaedic clinicians, specialist nurses and physiotherapists) working with orthopaedic patients at five NHS hospitals in England. It is anticipated that including the perspectives of both staff and patients, from a range of NHS hospitals will provide a more encompassing and detailed exploration of important QOL-related outcomes for patients.

### Participants and recruitment (selection of participants)

A sampling frame will be developed through collaboration with lead clinicians from the participating sites. Patients will be those who meet the 'Sample' eligibility criteria outlined in step A. Patients will be identified and invited to participate by orthopaedic clinicians, specialist nurses, physiotherapists or research nurses. The five sites are orthopaedic departments of varying sizes, with different orthopaedic specialities and are situated throughout England to offer geographical spread. A purposive sampling strategy will be adopted to ensure maximum variation according to age, gender, length of time since injury or condition onset, different lower limb conditions, trauma/condition severity and treatment received. Care will be taken to ensure that those who are requiring, undergoing or who have undergone reconstructive surgery for a lower limb condition, and have received care through the NHS at one of the participating

sites are included. To achieve maximum variation, we aim to recruit between 50 and 75 patient participants across the five sites (10–15 at each site) and 10 staff members (2 from each site); however, analysis will stop or continue beyond this if saturation has/has not been reached. Data collection will continue until saturation is reached and no new information is gained in the interviews/focus groups with regard to factors and outcomes surrounding patients' QOL.

Staff (orthopaedic clinicians, specialist nurses and physiotherapists) will be invited to participate in the study by either direct contact from the researcher or through the lead contact at each site who will be identified through the PROLLIT study's principal investigator's professional contacts. The anticipated start and end dates for data collection are November 2020–February 2021.

## Procedure

Patients will be invited to take part in either a one-to-one interview with a researcher (virtual or via telephone) or a focus group with a researcher and a maximum of seven other participants. Individual semistructured interviews or focus groups (approx. 4–8 people) will be undertaken with the patient participants and will be led by an experienced qualitative researcher. The interviews/focus groups will be arranged at a time most appropriate for the participants. A mix of focus groups and interviews for patients allows for flexibility in recruitment and scheduling the interviewers/focus groups. The group setting of the focus group may also provide an opportunity for patients to feel more relaxed and open, as well as enabling patients to bounce thoughts and discussion points off each other. Staff will be invited to take part in a one-to-one interview with the researcher either virtual or via telephone. Due to the limited time availability of working staff, it will not be practical to arrange focus groups.

It is anticipated that interviews will last between 30 and 60 minutes and focus groups between 1 and 2 hours. Topic guides will be used to guide the interviews/focus groups. These will be developed from a literature review of the area, our QES findings and the initial hypothesised conceptual framework developed from step A. The topic guides will be reviewed by experts in the field and our patient, public involvement group. The topic guide will use open-ended questions to explore key health-related QOL factors and other factors relevant to patients who have experienced a lower limb condition that requires or required reconstructive surgery.

Through the use of the topic guide and questioning, concept elicitation will be undertaken in the interview/focus group by asking the participants questions which explore their thoughts, attitudes and beliefs surrounding what is important to them with regard to QOL in relation to requiring, undergoing or after reconstructive surgery for a lower limb condition. Important factors may include physical, social and psychological well-being as well as job and lifestyle-related factors. The staff will be asked to discuss, from their perspective, what they perceive to be important treatment outcomes and goals for patients.

## Ethics and consent

We will ensure that participants have read the information sheet provided and have a full understanding of the study before consent is obtained. Participants will be invited to take part and will be reminded of their right to withdraw at any time. All participants (staff and patients) will be informed that participation is voluntary and that their involvement and responses will remain anonymous. Informed consent will be taken prior to each interview or focus group. Verbal consent will be obtained on an ongoing basis during interviews/focus groups. Participants will be reminded that the interviews/focus groups will be audio-recorded, quotations may be used and published, but that all identifiable information will be removed. All participants will be provided with a unique ID to maintain their anonymity.

## Analysis

Data will be analysed at the University of York by qualitative researchers (AS and HL), who will adopt an iterative approach to data analysis, involving regular discussion between the research team. The qualitative data will be recorded and transcribed verbatim with data analysis facilitated by the use of NVIVO. Quotations will be used where appropriate to provide evidence for the conclusions drawn when reporting the study's findings.

We have chosen to analyse our data using Framework analysis,[27] a method which allows for flexibility during the analysis process, as data can either be collected and then analysed, or analysed alongside data collection. The analysis will involve data being sifted, charted and sorted in accordance with key issues and themes. In this study, the focus will be on drawing out and grouping into key themes the important factors and outcomes relating to patients' self-reported QOL, as well as staff views on important treatment and outcome goals for patients. This will enable the research team to understand in greater depth, and from a psychological standpoint, the areas that are important to patients. To achieve this, our analysis will follow the five stages of framework analysis as outlined by Richie and Spencer:[27] (1) familiarisation; (2) identifying a thematic framework; (3) indexing; (4) charting and (5) mapping and interpretation. In stage 1, the researchers will become familiarised with and immersed in the transcripts by reading them through multiple times. Following this, coding of the transcripts will begin by identifying emerging themes related to patient's QOL. Coding will be deductively based on a priori themes from the QES findings where appropriate, while also inductively allowing for new themes to emerge which may influence the conceptual framework development. These will be influenced by and be complimentary to the findings from the QES. Multiple iterations of this step will enable the development of a thematic framework (stage 2). Indexing and charting (stages 3 and 4) will

begin whereby portions of the text that relate to a particular theme will be identified and charted according to the theme it corresponds to. Finally, in stage 5, mapping and interpretation involves analysis of the characteristics of each charted theme through graphical representation of the themes and how they relate to each other.[27] At the end of our analysis, the findings of the framework will be combined with those of the QES (step B) and used to develop the conceptual framework (step C).

### Step C: Conceptual framework development

An initial conceptual framework will be developed using findings from the QES and qualitative study. As previously discussed, findings will then be combined by the research team and a 'hypothesised conceptual framework' will be taken to a round table discussion with key stakeholders. Key stakeholders will include members of the project advisory panel, patient and public involvement group and research team.

Methodological guidance while developing the conceptual framework will be sought from guidelines and conceptual framework method papers as required.[28–31] The knowledge and experience of the advisory panel and the patient and public involvement group will also be vital for the final refinement of the conceptual framework in step C and they will be invited to work closely alongside the research team at this stage. The final conceptual framework will represent what is important to patients with regard to social interactions, employment, perceived health and QOL after injury and throughout recovery after lower limb reconstructive surgery.

### ADVISORY PANEL

An advisory panel has been convened, consisting of orthopaedic surgeons, physiotherapists and methodologists to provide support, opinion and insight at key times during the project and to support recruitment to the study. The group will be contacted through a variety of methods such as emails, face-to-face meetings, telephone and video conferencing.

### PATIENT AND PUBLIC INVOLVEMENT

A patient and public involvement group will be engaged throughout all stages of the study from design to reviewing the final conceptual framework. To maximise the diversity of the group, we will aim to recruit people from a range of relevant patient support and clinical groups such as physiotherapy groups at each of the study sites. Patient and public involvement will include, but is not limited to, reviewing documentation for the qualitative study as well as sense checking the domains of importance identified from the QES and the final conceptual framework. This will ensure that everything is appropriate and comprehensible from a patient's perspective. Patients will not be involved in the recruitment or conduct of the study. The results of the study will be disseminated to the patient and public involvement group via email or shared at a regular meeting.

### ETHICS AND DISSEMINATION

Only step B requires ethical approval which has been given favourable opinion from South Central—Berkshire Research Ethics committee (REF:20/SC/0114). All necessary local research governance approvals will be obtained at each of the five study sites prior to data collection. The interviews/focus groups will not impact on the care of the patients or the staff professionally. All participants will be provided with written information about the study and informed consent will be obtained. Step A is registered on the international prospective register of systematic reviews (PROSPERO) database (CRD number: CRD42019139587) and step B is registered on the International Standard Randomised Controlled Trial Number Registry (ISRCTN75201623). We intend to publish the QES findings and the findings from the qualitative study in a peer-reviewed journal, disseminate at relevant conferences and provide feedback on the findings to participants who are interested and other patient groups and organisations.

### RISKS AND MITIGATIONS

Potential risks to the undertaking of this research pertain mainly to the recruitment of participants for step B. Due to COVID-19, it may be harder to recruit patients due to reduced contact between hospital staff and patients which may make it difficult to achieve our desired sample size or to obtain a representative sample of patients. To mitigate this, the researchers will closely monitor recruitment and if necessary ask those identifying patients to actively recruit certain types of patients. To mitigate the risk of COVID-19 to participants and researchers, all interviews and focus groups will now be conducted either virtually via an online platform or via the telephone.

**Acknowledgements** The authors would like to thank the advisory panel and the advisory patient and public involvement group who provided insight into defining the population of interest, refining the study design and providing feedback on key materials for step B.

**Collaborators** The PROLLIT study collaborators: Joy Adamson, PhD (The University of York), Georgina Jones, PhD (Leeds Beckett University), Kim Cocks, PhD (KCStats Consultancy), Joel Gagnier, PhD (University of Michigan), Paul Harwood, MBChB (Leeds Teaching Hospitals NHS Trust), David Ferguson, MBChB (South Tees Hospitals, NHS Trust), Reggie Hamdy, MBChB (McGill University) and Nando Ferriera, PhD (Stellenbosch University).

**Contributors** CM, CH and HS were involved in the study conception. CM, HS, CH, HL and AS designed the study. MH developed the search strategy for the QES. HL drafted the manuscript with input from AS, CM, HS, MH and CH. The PROLLIT collaborators provided methodological and clinical advice. All authors and the PROLLIT study collaborators reviewed and approved the final manuscript.

**Funding** This project is funded through Research Capability funding from Hull University Teaching Hospitals NHS Trust.

**Competing interests** CM reports grants from Hull University Teaching Hospitals NHS Trust, during the conduct of the study; grants from British Orthopaedic

Association, outside the submitted work. HS reports grants from Health & Technology Assessment, during the conduct of the study; grants from B.Braun, personal fees from Biocomposites and personal fees from Orthofix, outside the submitted work. MH, HL, AS and CH have nothing to disclose.

**Patient consent for publication** Not required.

**Provenance and peer review** Not commissioned; externally peer reviewed.

**ORCID iDs**
Heather Leggett http://orcid.org/0000-0001-8708-9842
Arabella Scantlebury http://orcid.org/0000-0003-3518-2740
Catriona McDaid http://orcid.org/0000-0002-3751-7260

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
