## [Reviewer comments · BMJ Open]

ARTICLE DETAILS

TITLE (PROVISIONAL)	Quality of life following a lower limb reconstructive procedure: A protocol for the development of a conceptual framework
AUTHORS	Leggett, Heather; Scantlebury, Arabella; Sharma, Hemant; Hewitt, Catherine; Harden, Melissa; McDaid, Catriona

VERSION 1 – REVIEW

REVIEWER	F. Usueli Humanitas San Pio X, Italy, Milan
REVIEW RETURNED	02-Jun-2020

GENERAL COMMENTS	Do not use the first person (We...)prefer passive voice Abstract: very well reported also if it's not clear the aim of the study. Methods: how the sample size was measured? Introducion: is not totally clear what do you mean for lower limb reconstruction. this is really confusing. Introduction is quite long.please shortened if possibile methods: was a power analysis performed?Inclusion and exclusion criteria are missing what this article will bring to the current literature and to the daily clinical practice? please add
---

REVIEWER	Jiao Jiao Li University of Sydney, Australia
REVIEW RETURNED	05-Jun-2020

GENERAL COMMENTS	This protocol is well constructed and aims to develop a useful tool for the evaluation of quality of life in lower limb amputation and reconstruction. Cited references are relevant and up to date. Some comments to improve the protocol are as follows: - Introduction, paragraph 2 – The Q-TFA is also a commonly used subjective outcome measure for transfemoral lower limb amputations and should probably be mentioned with the others. Also, some functional outcome measures that have been applied to lower limb amputations may be mentioned, such as 6 minute walk test and timed up and go test.- Introduction – There should be more clarification surrounding the specific gaps which PROLLIT is aiming to address, i.e. specify what is missing from the existing outcome measures currently applied for lower limb amputations and what PROLLIT will be able to provide that differs from all of these tools.
---

	- Step C, conceptual framework development – It would be good to clarify how the membership for the “patient and public involvement and engagement group” will be determined. Will members be selected based on diversity, impact of the condition, a mixture, etc. and how to ensure that the selection of members is representative of the target group. This is important to clarify since the tool is centred around patient involvement and patient-based experiences. - There should be a discussion section that incorporates the following points:  1. Some details should be given around the practicality of conducting this protocol – what are the approximate time frames required, are there potential risks that may delay the study or prevent its successful completion, and are there contingency plans available/considered. 2. Limitations of this study protocol and how this may affect the outcomes of the study should be described. 3. There should be an outline for how the developed tool will be evaluated and validated – are there plans for the same group or its collaborators to conduct a clinical study for evaluation?
--	--

REVIEWER	Nabil EBraheim University of Toledo Medical Center, United States
REVIEW RETURNED	16-Jun-2020

GENERAL COMMENTS	This paper strives to identify a need and propose a framework for a multi-phase investigation to better evaluate outcomes following lower limb reconstructive procedures. As an endeavor, it certainly has merit, and the paper is well written. As it currently stands, however, there are no data or tangible results from the study. This will be a valuable addition to the wider body of literature as the project evolves, but whether the protocol in and of itself is publishable may be questionable. As for grammar and substance, there aren't any qualms.
--

REVIEWER	Sander L. Hitzig Sunnybrook Research Institute, Canada
REVIEW RETURNED	04-Sep-2020

GENERAL COMMENTS	This is a well written protocol and it is important to ensure that there are adequate patient-reported outcome measures (PROMs) to evaluate the impact of lower limb conditions. I have some points of consideration for the authors listed below: In the abstract, the authors write the impact of a patient's condition (and treatment) on quality of life and health related quality of life. I think given the focus of the condition on quality of life, the health-related quality of life part of the sentence can be dropped. I recognize that the study may or may not develop a new PROM dependent on the quality of the existing evidence but wondering if they do decide to develop a new measure, should they provide some indication of an established PROM guideline framework they intend to follow? (e.g., Haywood et al. – ref pasted below as one possible guideline framework). Haywood KL, de Wit M, Staniszevska S, Morel T, Salek S. Developing Patient-Reported and Relevant Outcome Measures. In: Facey KM, Single ANV, Ploug Hansen H, eds. Patient Involvement
---

	in Health Technology Assessment. Singapore: Springer Nature; 2017. I am wondering why the authors are only including qualitative studies in their Step A. While I agree the inclusion of qualitative studies are important, the authors highlight in their introduction that there are quantitative measures used in their target population, which capture QoL even if they aren't condition specific. Wouldn't it be relevant to include these types of studies to ensure that they have as much relevant information regarding the QoL domains for this population? Relatedly, a line should be provided in the introduction about the limitations of some of the measures listed (e.g., SF-36, EQ-5DL). They state on lines 24-25 that these tools have been effective in capturing and reporting on the quality of life in patients. Although it is implicit, perhaps a line about the need for a condition specific measure is warranted? The authors may intend to do this but if not, they should follow a standardized protocol if they are going to de-duplicate in EndNote, such as Bramer et al.'s method https://www.ncbi.nlm.nih.gov/pmc/articles/PMC4915647/ Although the introduction is well-written, I don't have a great sense of the characteristics / impact of limb conditions in the introduction. The authors state it negatively influences QoL but some nuance should be provided. For instance, a sentence should be provided before line 9 such as: "Persons with lower limb conditions typically have mobility impairments, pain, other comorbidities, which reduces opportunities for social engagement. (refs). Therefore, it is not surprising that the experience of living with a lower limb condition, as well as the impact of initial trauma..." Are the authors also examining persons with dysvascular conditions who may undergo lower limb reconstructive surgery? Line 40 – Minor consideration: I would rewrite '...patients are still being underused' to '...patients are not being fully engaged..' or something along those lines.
--	--

VERSION 1 – AUTHOR RESPONSE

Reviewer	Comments	Response
Reviewer: 1	Do not use the first person (We...)prefer passive voice	Thank you for this suggestion. We respect that there are different perspectives on this but we have tried to follow the journal guidance on house style which requests that articles are written in an active style. However, we are happy to respond to further editorial guidance on this.
	Abstract: very well reported also if it's not clear the aim of the study. Methods: how the sample size was measured?	Aim: Thank you, we have revised the abstract to make the aim of the study clearer.

		Methods: the protocol is for the development of a conceptual framework, which comprises a qualitative evidence synthesis, qualitative study and roundtable discussion. As the study does not include any quantitative methods it is not appropriate to include a sample size calculation. For the qualitative study, a detailed description of our sampling strategy, which uses an established qualitative method (purposive sampling) is outlined.
	Introducion: is not totally clear what do you mean for lower limb reconstruction. this is really confusing. Introduction is quite long.please shortened if possibile	We have added some information to clarify this: Limb reconstruction is the orthopaedic subspecialty, treating sequel of complex trauma and congenital conditions; deformity correction, nonunion, malunion, bone infections, limb lengthening and complex trauma.
	methods: was a power analysis performed?Inclusion and exclusion criteria are missing	A power analysis was not performed as this is not appropriate for qualitative research. As stated above, our sampling criteria are clearly outlined for both patients and orthopaedic staff using established qualitative techniques. Clarification over the inclusion criteria has been added to the systematic review methods. We have also provided some further clarification on who will be recruited to the qualitative study.
	what this article will bring to the current literature and to the daily clinical practice? please add	Our introduction includes an overview of why PROMS are important and how they can improve our understanding of the impact of treatment on patient's quality of life as well as improve care. We also outline gaps in existing evidence and specifically, why there may be a need to develop a new PROM. We have however added some

		additional text in the introduction to help clarify this point.
Reviewer: 2	This protocol is well constructed and aims to develop a useful tool for the evaluation of quality of life in lower limb amputation and reconstruction. Cited references are relevant and up to date. Some comments to improve the protocol are as follows: - Introduction, paragraph 2 – The Q-TFA is also a commonly used subjective outcome measure for transfemoral lower limb amputations and should probably be mentioned with the others. Also, some functional outcome measures that have been applied to lower limb amputations may be mentioned, such as 6 minute walk test and timed up and go test.	Our population of interest, does not include individuals who are waiting for, or who have recently undergone amputation. We have clarified this in our methods section.
	- Introduction – There should be more clarification surrounding the specific gaps which PROLLIT is aiming to address, i.e. specify what is missing from the existing outcome measures currently applied for lower limb amputations and what PROLLIT will be able to provide that differs from all of these tools.	The aim of this first stage of the PROLLIT study is to establish what is important to patients and then to establish whether there are indeed any gaps in current measures. An outcome of this study may be that a further outcome measure is not required.
	- Step C, conceptual framework development – It would be good to clarify how the membership for the “patient and public involvement and engagement group” will be determined. Will members be selected based on diversity, impact of the condition, a mixture, etc. and how to ensure that the selection of members is representative of the target group. This is important to clarify since the tool is centred around patient involvement and patient-based experiences.	We agree that a diverse PPI group is important. To maximise the diversity of the group we will aim to try and engage people through a range of sources such as physiotherapy and support groups. We have added information into the “Advisory Panel and Patient and Public Involvement and Engagement” section of the paper. However we do not feel it is appropriate to have a ‘sampling frame’ approach as we do in the research element

	- There should be a discussion section that incorporates the following points:  1. Some details should be given around the practicality of conducting this protocol – what are the approximate time frames required, are there potential risks that may delay the study or prevent its successful completion, and are there contingency plans available/considered. 2. Limitations of this study protocol and how this may affect the outcomes of the study should be described. 3. There should be an outline for how the developed tool will be evaluated and validated – are there plans for the same group or its collaborators to conduct a clinical study for evaluation? 	1 &2. Thanks for this suggestion. The time frames are listed in Step B. page 8. We have added a risks and mitigations section to the script, see pq 10. 3. Thank-you for your interest in the third phase of the study and as a research team we would hope to see the project through to completion. We did consider whether we should do a protocol paper of all three phases. However, each phase uses very different methods and it would not be feasible to fit it all into a single paper within expected word counts. Also, whether we develop a new PROM, will depend on the quality of existing evidence and the outcome of the work we propose to do in this first phase of the wider PROLLIT study and we do not want to make assumptions about what the findings will be.
Reviewer: 3	This paper strives to identify a need and propose a framework for a multi-phase investigation to better evaluate outcomes following lower limb reconstructive procedures. As an endeavor, it certainly has merit, and the paper is well written. As it currently stands, however, there are no data or tangible results from the study. This will be a valuable addition to the wider body of literature as the project evolves, but whether the protocol in and of itself is publishable may be questionable. As for grammar and substance, there aren't any qualms.	We are pleased that the reviewer considers the manuscript to be well written and to have merit. As this is a protocol paper, there are no results. We have reported the plans for the study as per BMJ Opens guidance for publishing protocols.
Reviewer: 4	This is a well written protocol and it is important to ensure that there are adequate patient-reported outcome measures (PROMs) to evaluate the impact of lower limb conditions. I have	Thank you for this suggestion. We have removed the health related reference.

	some points of consideration for the authors listed below: In the abstract, the authors write the impact of a patient's condition (and treatment) on quality of life and health related quality of life. I think given the focus of the condition on quality of life, the health-related quality of life part of the sentence can be dropped.	
	I recognize that the study may or may not develop a new PROM dependent on the quality of the existing evidence but wondering if they do decide to develop a new measure, should they provide some indication of an established PROM guideline framework they intend to follow? (e.g., Haywood et al. – ref pasted below as one possible guideline framework). Haywood KL, de Wit M, Staniszevska S, Morel T, Salek S. Developing Patient-Reported and Relevant Outcome Measures. In: Facey KM, Single ANV, Ploug Hansen H, eds. Patient Involvement in Health Technology Assessment. Singapore: Springer Nature; 2017.	We thank the reviewer for the pointer to this reference, it looks very useful for the potential next step of PROM development. However, as recognised by the reviewer, , whether this next step is required is dependent on the findings of phase 1. We therefore feel it would be premature and out of scope of this manuscript to address this.
	I am wondering why the authors are only including qualitative studies in their Step A. While I agree the inclusion of qualitative studies are important, the authors highlight in their introduction that there are quantitative measures used in their target population, which capture QoL even if they aren't condition specific. Wouldn't it be relevant to include these types of studies to ensure that they have as much relevant information regarding the QoL domains for this population? Relatedly, a line	Thank you for this suggestion. This was a source of considerable discussion at the outset of the study. We decided to not include any quantitative studies because it was felt that whilst a quantitative study will tell us what domains that tool/paper explored it won't tell us if they were important or how they were important to patients. This is the element of QoL domains we were interested in and is captured by the qualitative papers.

	should be provided in the introduction about the limitations of some of the measures listed (e.g., SF-36, EQ-5DL). They state on lines 24-25 that these tools have been effective in capturing and reporting on the quality of life in patients. Although it is implicit, perhaps a line about the need for a condition specific measure is warranted?	Regarding the comment about limitations of the measures we have edited the sentence about them being effective as we agree that is making an assumption. We have also added a further sentence to be more explicit about the need to a condition-specific measure.
	The authors may intend to do this but if not, they should follow a standardized protocol if they are going to de-duplicate in EndNote, such as Bramer et al.'s method https://www.ncbi.nlm.nih.gov/pmc/articles/PMC4915647/	Thank you for this suggestion, we are aware of the deduplication method by Bramer et al. however we used our own tried and tested deduplication algorithms in EndNote, developed by information specialists at the Centre for Reviews and Dissemination at the University of York.
	Although the introduction is well-written, I don't have a great sense of the characteristics / impact of limb conditions in the introduction. The authors state it negatively influences QoL but some nuance should be provided. For instance, a sentence should be provided before line 9 such as: "Persons with lower limb conditions typically have mobility impairments, pain, other comorbidities, which reduces opportunities for social engagement. (refs). Therefore, it is not surprising that the experience of living with a lower limb condition, as well as the impact of initial trauma..."	Thank for you this suggestion, we agree this would be valuable to add in and have done so.
	Are the authors also examining persons with dysvascular conditions who may undergo lower limb reconstructive surgery?	We will be focussing on limb reconstruction in post trauma conditions. Dysvascular conditions are not the focus of this study
	Line 40 – Minor consideration: I would rewrite '...patients are still being underused' to '..patients are not being fully engaged..' or something along those lines.	Thank you for this suggestion. We have reworded using your phrasing.

VERSION 2 – REVIEW

REVIEWER	Jiao Jiao Li University of Technology Sydney, Australia
REVIEW RETURNED	07-Oct-2020

GENERAL COMMENTS	The authors have addressed my previous comments. I have no new comments to add.
---

REVIEWER	Sander L. Hitzig Sunnybrook Research Institute, Canada
REVIEW RETURNED	19-Oct-2020

GENERAL COMMENTS	I am satisfied by the responses by the authors and the revised manuscript is much improved as a result. A small note is that they introduce the acronym (QOL) but then write out quality of life in several instances past that.
--

VERSION 2 – AUTHOR RESPONSE

Dear Reviewers,

Thank you for your comments. We are glad our revisions have resolved your concerns of the previous submission. Thank you to reviewer 2 for highlighting our error of continuing to type of quality of life after using the QOL acronym. This has now been resolved, apart from in one instance where quality of life was at the beginning of a sentence.

Editor, apologies for not including the name of the ethics committee in the abstract. This has now been amended and the abstract modified to meet the word limit. The study was approved by one ethics committee which is named in the ethics and dissemination section in the main text so no change has been made here.

With best wishes,

The authors.